# A Dataset of Students' Mental Health and Help-Seeking Behaviors in a Multicultural Environment

**Minh-Hoang Nguyen [1] , Manh-Toan Ho [2,3,]\* , Quynh-Yen T. Nguyen [4,]\* and Quan-Hoang Vuong [2,3]**

[1] International Cooperation Policy, Graduate School of Asia Pacific Studies, Ritsumeikan Asia Pacific University, Beppu, Oita 874-8577, Japan
[2] Center for Interdisciplinary Social Research, Phenikaa University, Ha Dong district, Hanoi 100803, Vietnam
[3] Faculty of Economics and Finance, Phenikaa University, Ha Dong district, Hanoi 100803, Vietnam
[4] College of Asia Pacific Studies, Ritsumeikan Asia Pacific University, Beppu, Oita 874-8577, Japan
\* Correspondence: toan.homanh@phenikaa-uni.edu.vn (M.-T.H.); thiqng17@apu.ac.jp (Q.-Y.T.N.)

**Abstract:** University students, especially international students, possess a higher risk of mental health problems than the general population. However, the literature regarding the prevalence and determinants of mental health problems as well as help-seeking behaviors of international and domestic students in Japan seems to be limited. This dataset contains 268 records of depression, acculturative stress, social connectedness, and help-seeking behaviors reported by international and domestic students at an international university in Japan. One of the main findings that can be drawn from this dataset is how the level of social connectedness and acculturative stress are predictive of the reported depression among international as well as domestic students. The dataset is expected to provide reliable materials for further study of cross-cultural public health studies and policy-making in higher education.

**Dataset:** The dataset is submitted as a supplement to this manuscript.

**Dataset License:** CC-BY

**Keywords:** higher education; international student; depression; acculturative stress; social connectedness; suicidal ideation; help-seeking; Japan; PHQ-9; ASSIS; bayesvl

## 1. Summary

This paper presents a comprehensive dataset of mental health condition (depression, acculturative stress, social connectedness, and suicidal ideation) and help-seeking behaviors of international and domestic students in an international university in Japan. The university is famous for the multicultural environment with 50% of students and faculties being international [1,2].

The questionnaire used for the survey was designed using elements from 4 standard measurements: Patient Health Questionnaire (PHQ-9), Acculturative Stress Scale for International Students (ASSIS), Social Connectedness Scale (SCS), and General Health Help-Seeking Questionnaire (GHSQ). The survey strictly conformed the World Medical Association (WMA) Declaration of Helsinki and was permitted by the Ethical Committee Board of Ritsumeikan Asia Pacific University (APU) after an internal review. The survey was distributed from October to December 2018. Google Forms was used to distribute the questionnaire due to its familiarity with students (Sample link: https://forms.gle/zAgByNfHN1LNfnAz6). The survey strictly conformed to APU regulations of informing participants

about consent text and purpose of the research at the beginning of the survey. The total response rate was 40.05% (268/669).

The current dataset is one of the first comparative datasets giving insights about mental health conditions and help-seeking behaviors of international and domestic students in Japan. Some findings from this dataset have been used in several publications [3,4]. In addition to other aspects of public health such as health insurance [5] and health care quality [6], intensive explanations and raw records in this data descriptor are expected to provide reliable resources to researchers for cross-cultural public health studies as well as for mental health policy-making.

First, in the Data Description section, we explain the entire set of coded variables and introduce some potential research questions and hypotheses that can be investigated based on the dataset. Then, in the Methods section, we give some examples of data analysis. Finally, the paper is wrapped up by showing certain limitations and implications of the dataset.

## 2. Data Description

The dataset comprises of 268 records from both international and domestic students in an international university in Japan (see Supplementary Material). This dataset is used to examine the mental health conditions and help-seeking behaviors of international and domestic students in a multicultural environment.

All answers included in the questionnaire are multiple-choice questions. The questions were divided into three main groups: (1) socio-demographic information, (2) mental health conditions (depression, suicidal ideation, acculturative stress, social connectedness), (3) help-seeking behaviors. In total, there are 25 categorical variables (see Table 1) and 26 numerical variables (see Table 2) created.

### 2.1. Socio-Demographic Information

In this sub-section, variables regarding student's home country, gender, academic level, length of stay, languages proficiency, religion, and whether being in an intimate relationship or not are presented and explained. The number of international participants accounted for 75%, higher than domestic respondents with 25%. Students from Southeast Asia with 45.52% occupied the largest proportion of respondents while students from Japan ranks second with 25.75%.

**Table 1.** Categorical variables.

| Coded Name | Explanation | Unit | Frequency | Proportion |
|---|---|---|---|---|
| inter_dom | Types of students: International student (Inter) or domestic student (Dom) | Inter<br>Dom | 201<br>67 | 75.00%<br>25.00% |
| Region | Regions where students originally come from: Japan (JAP), East Asia excluding Japan (EA), South Asia (EA), South East Asia (SEA) or other regions (Others) | JAP<br>SA<br>EA<br>SEA<br>Others | 69<br>18<br>48<br>122<br>11 | 25.75%<br>6.72%<br>17.91%<br>45.52%<br>4.10% |
| Gender | Gender of students: Male or Female | Male<br>Female | 98<br>170 | 36.57%<br>63.43% |
| Academic | The current academic level of students: Undergraduate (Under) or Graduate School (Grad) | Under<br>Grad | 247<br>21 | 92.16%<br>7.84% |
| Stay_Cate | How long students have been at the university: 1 year (Short), 2–3 years (Medium) or at least 4 years (Long) | Short<br>Medium<br>Long | 115<br>121<br>32 | 42.91%<br>45.15%<br>11.94% |
| Japanese_cate | Self-evaluation scale ranging from 1 to 5 regarding Japanese proficiency: High (4 to 5), Medium (3) or Low (1 to 2) | High<br>Average<br>Low | 87<br>89<br>92 | 32.46%<br>33.21%<br>34.33% |
| English_cate | Self-evaluation scale ranging from 1 to 5 regarding English proficiency: High (4 to 5), Medium (3) or Low (1 to 2) | High<br>Average<br>Low | 166<br>80<br>22 | 61.94%<br>29.85%<br>8.21% |
| Intimate | Whether students have an intimate partner or not | Yes<br>No | 103<br>157 | 38.43%<br>58.58% |

**Table 1.** *Cont.*

| Coded Name | Explanation | Unit | Frequency | Proportion |
|---|---|---|---|---|
| Religion | Whether students are religious or not | Yes<br>No | 91<br>177 | 33.96%<br>66.04% |
| Suicide | Whether students have suicidal Ideation in the last 2 weeks or not (based on a question in PHQ-9) | 61<br>207 | 61<br>207 | 22.76%<br>77.24% |
| Dep | Whether students reported having depressive symptoms based on PHQ-9 criteria | Yes<br>No | 96<br>172 | 35.82%<br>64.18% |
| DepType | Types of depressive disorder based on PHQ-9 criteria: Major depressive disorder (Major), Other depressive disorder (Other), and no depressive disorder (No) | Major<br>Other<br>No | 42<br>54<br>172 | 15.67%<br>20.15%<br>64.18% |
| DepSev | The severity of depressive disorder based on PHQ-9 criteria: Minimal depression (Min), Mild depression (Mild), Moderate depression (Mod), Moderately severe depression (ModSev), Severe depression (Sev) | Min<br>Mild<br>Mod<br>ModSev<br>Sev | 65<br>107<br>73<br>15<br>8 | 24.25%<br>39.93%<br>27.24%<br>5.60%<br>2.99% |
| Partner_bi | Whether students are willing to seek help from an intimate partner when they encounter emotional difficulties | Yes<br>No | 145<br>123 | 54.10%<br>45.90% |
| Friends_bi | Whether students are willing to seek help from friends when they encounter emotional difficulties | Yes<br>No | 128<br>140 | 47.76%<br>52.24% |
| Parents_bi | Whether students are willing to seek help from parents when they encounter emotional difficulties | Yes<br>No | 137<br>131 | 51.12%<br>48.88% |
| Relative_bi | Whether students are willing to seek help from relatives when they encounter emotional difficulties | Yes<br>No | 66<br>202 | 24.63%<br>75.37% |
| Professional_bi | Whether students are willing to seek help from professionals when they encounter emotional difficulties | Yes<br>No | 61<br>207 | 22.76%<br>77.24% |
| Phone_bi | Whether students are willing to seek help from phone helpline when they encounter emotional difficulties | Yes<br>No | 30<br>238 | 11.19%<br>88.81% |
| Doctor_bi | Whether students are willing to seek help from a doctor when they encounter emotional difficulties | Yes<br>No | 46<br>222 | 17.16%<br>82.84% |
| religion_bi | Whether students are willing to seek help from a religious leader when they encounter emotional difficulties | Yes<br>No | 19<br>249 | 7.09%<br>92.91% |
| Alone_bi | Whether students are willing to solve problems by themselves | Yes<br>No | 65<br>203 | 24.25%<br>75.75% |
| Internet_bi | Whether students are willing to seek help from the Internet when they encounter emotional difficulties | Yes<br>No | 45<br>223 | 16.79%<br>83.21% |
| Others_bi | Whether students are willing to seek help from other sources not listed above when they encounter emotional difficulties | Yes<br>No | 21<br>247 | 7.84%<br>92.16% |

Figure 1 gives information about the age distribution collected from the questionnaire. Participants' age ranges from 17 to 31, with the mean being 20.87. It can be seen that the graph is a right-skewed distribution. The reason for this is because graduate students (7.84%) who participated in the survey are relatively older than undergraduate students (92.16%).

Other variables such as gender, length of stay, language proficiency, and religion were also collected to the dataset. Female participants accounted for 63.43% while this proportion for male students was 36.57%, and until the reported time, most participants had been in this university for 1 to 3 years.

Regarding language proficiency, students were asked to self-evaluate their English and Japanese ability on a scale from 1 to 5. Majority of participants rate themselves 4 or 5, equivalent to high proficiency, for English proficiency (61.94%), while Japanese language evaluation spread equally from low to high proficiency.

Most students reporting to the survey did not consider themselves religious (66.04%). Approximately 60% of students said they did not have an intimate partner (several participants did not respond if they were in an intimate relationship).

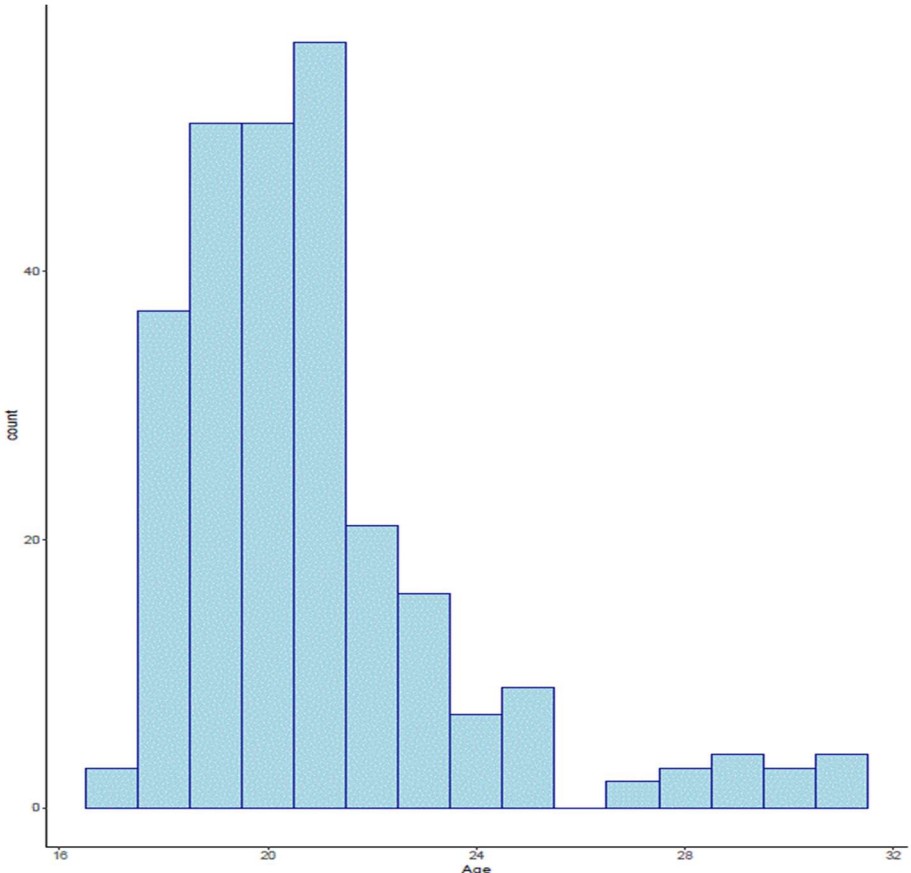

**Figure 1.** Age distribution of respondents.

*2.2. Mental Health Conditions*

To measure Depression, we employed the Patient Health Questionnaire PHQ-9, which consists of nine questions to examine the depression symptoms as well as depression severity [7,8]. The PHQ-9 is a commonly used questionnaire to screen for depression in many different populations and medical settings [9–12]. Students were asked to report the frequency of nine symptoms in the last two weeks based on a 4-point Likert scale ranging from 0 (not at all) to 3 (nearly every day). Based on the PHQ-9, four categorical variables ("Suicide", "Dep", "DepType", "DevSev") and one continuous variable ("ToDep") were created. The measured Cronbach alpha for the international and domestic dataset was 0.81 and 0.80, respectively [3], while the validity of the question was confirmed by other mental health studies [12,13].

The level of Acculturation was measured by Acculturative Stress Scale for International Students (ASSIS) [14]. The scale covers seven categories, including Perceived Discrimination, Homesickness, Perceived Hatred, Fear, Culture Shock, Guilt, and Miscellaneous. Participants reported by scoring on a 5-point Likert scale from 1 (strongly disagree) to 5 (strongly agree). There are eight numerical variables created according to the ASSIS questions ("APD", "AHome", "APH", "AFear", "ACS", "AGuilt", "AMiscell", "ToAS". The internal reliability of the questionnaire was 0.95, and the questionnaire's validity was also supported by the following studies [15–17].

**Table 2.** Numerical variables.

| Coded Name | Explanation | Unit | Mean | Min | Max | Standard Deviation |
|---|---|---|---|---|---|---|
| Age | Current age of students | Age | 20.87 | 17 | 31 | 2.77 |
| Stay | How long they have been in the university | Year | 2.15 | 1 | 10 | 1.33 |
| Jap | Self-evaluation scale ranging from 1 to 5 regarding Japanese proficiency | Count | 3.10 | 1 | 5 | 1.31 |
| Eng | Self-evaluation scale ranging from 1 to 5 regarding English proficiency | Count | 3.65 | 1 | 5 | 0.88 |
| ToDep | Total score of depression measured by PHQ-9 | Count | 8.19 | 0 | 25 | 4.95 |
| ToSC | Total social connectedness measured by SCS | Count | 37.47 | 8 | 48 | 9.23 |
| APD | The total score of perceived discrimination measured by ASISS questionnaire | Count | 15.41 | 8 | 39 | 6.17 |
| AHome | The total score of homesickness measured by ASISS questionnaire | Count | 9.61 | 4 | 20 | 4.01 |
| APH | The total score of perceived hatred measured by ASISS questionnaire | Count | 9.14 | 5 | 25 | 4.19 |
| AFear | The total score of fear measured by ASISS questionnaire | Count | 7.26 | 4 | 17 | 3.11 |
| ACS | The total score of culture shock measured by ASISS questionnaire | Count | 6.06 | 3 | 13 | 2.60 |
| AGuilt | The total score of guilt measured by ASISS questionnaire | Count | 3.78 | 2 | 10 | 1.91 |
| AMiscell | The total score of miscellaneous measured by ASISS questionnaire | Count | 21.12 | 10 | 47 | 7.40 |
| ToAS | Total score of Acculturative Stress | Count | 72.38 | 36 | 145 | 22.64 |
| Partner | Willingness to seek help from an intimate partner when students encounter emotional difficulties | Count | 4.32 | 1 | 7 | 2.23 |
| Friends | Willingness to seek help from friends when students encounter emotional difficulties | Count | 4.06 | 1 | 7 | 1.94 |
| Parents | Willingness to seek help from parents when students encounter emotional difficulties | Count | 4.37 | 1 | 7 | 2.07 |
| Relative | Willingness to seek help from relatives or family members when students encounter emotional difficulties | Count | 3.07 | 1 | 7 | 1.85 |
| Profess | Willingness to seek help from professionals when students encounter emotional difficulties | Count | 2.95 | 1 | 7 | 1.83 |
| Phone | Willingness to seek help from phone line when students encounter emotional difficulties | Count | 2.29 | 1 | 7 | 1.53 |
| Doctor | Willingness to seek help from doctors when students encounter emotional difficulties | Count | 2.67 | 1 | 7 | 1.71 |
| Reli | Willingness to seek help from religious leader when students encounter emotional difficulties | Count | 1.92 | 1 | 7 | 1.41 |
| Alone | Willingness to solve problems by themselves when students encounter emotional difficulties | Count | 2.94 | 1 | 7 | 2.03 |
| Others | Willingness to seek help from other sources not listed above when students encounter emotional difficulties | Count | 2.14 | 1 | 7 | 1.50 |
| Internet | Willingness to seek help from the internet when students encounter emotional difficulties | Count | 3.02 | 1 | 7 | 1.64 |

Figure 2 shows the boxplot between types of depressive symptoms and types of acculturative stress among international and domestic students. Students reporting suffering from major depressive disorder seemed to receive higher scores in all kinds of acculturative stress than those who have no or other depressive disorders. Notably, in all cases, the score of international students was relatively more substantial than that of domestic students. Moreover, both international and domestic students without depressive disorders might also be less likely to suffer from acculturative stress.

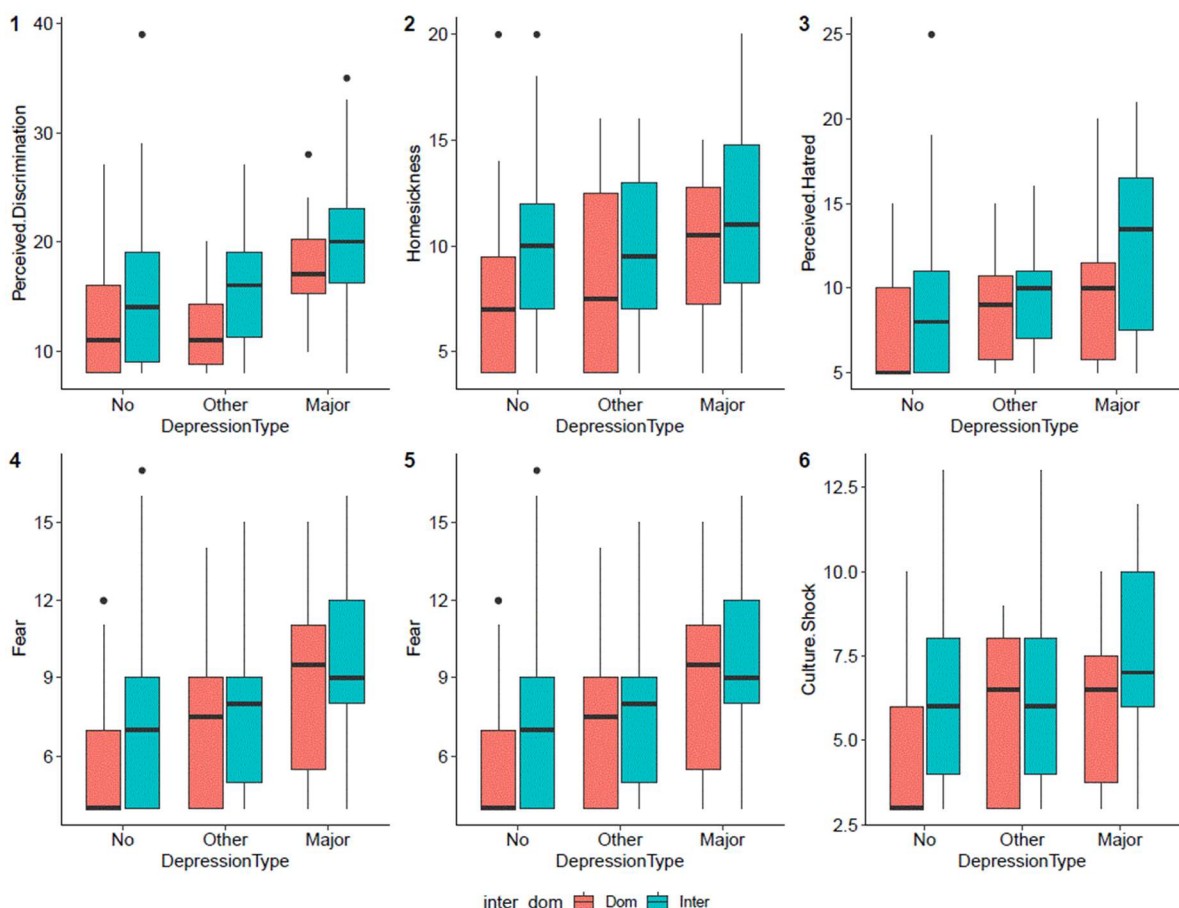

**Figure 2.** Different types of acculturative stress according to the type of depressive disorder.

The Social Connectedness Scale (SCS) developed by Lee and Robins was the tool used to measure individual emotional distance or connectedness among themselves and other people [18]. The questionnaire consisted of eight questions in which each of them were rated on the 6-point Likert scale ranging from 1 (Strongly Disagree) to 6 (Strongly Agree). One continuous variable was created from the questionnaire ("ToSC"). The coefficient alpha of SCS was 0.95, and the validity of the questionnaire was confirmed by [19,20].

The boxplot between students' language proficiency and Acculturation stress and Social Connectedness is presented in Figure 3. Among international and domestic students, higher fluency in Japanese might result in less acculturative stress, while the score of total connectedness remained almost similar in all levels of Japanese proficiency. As for English proficiency, there were no clear correlational tendencies between language proficiency and acculturative stress as well as social connectedness.

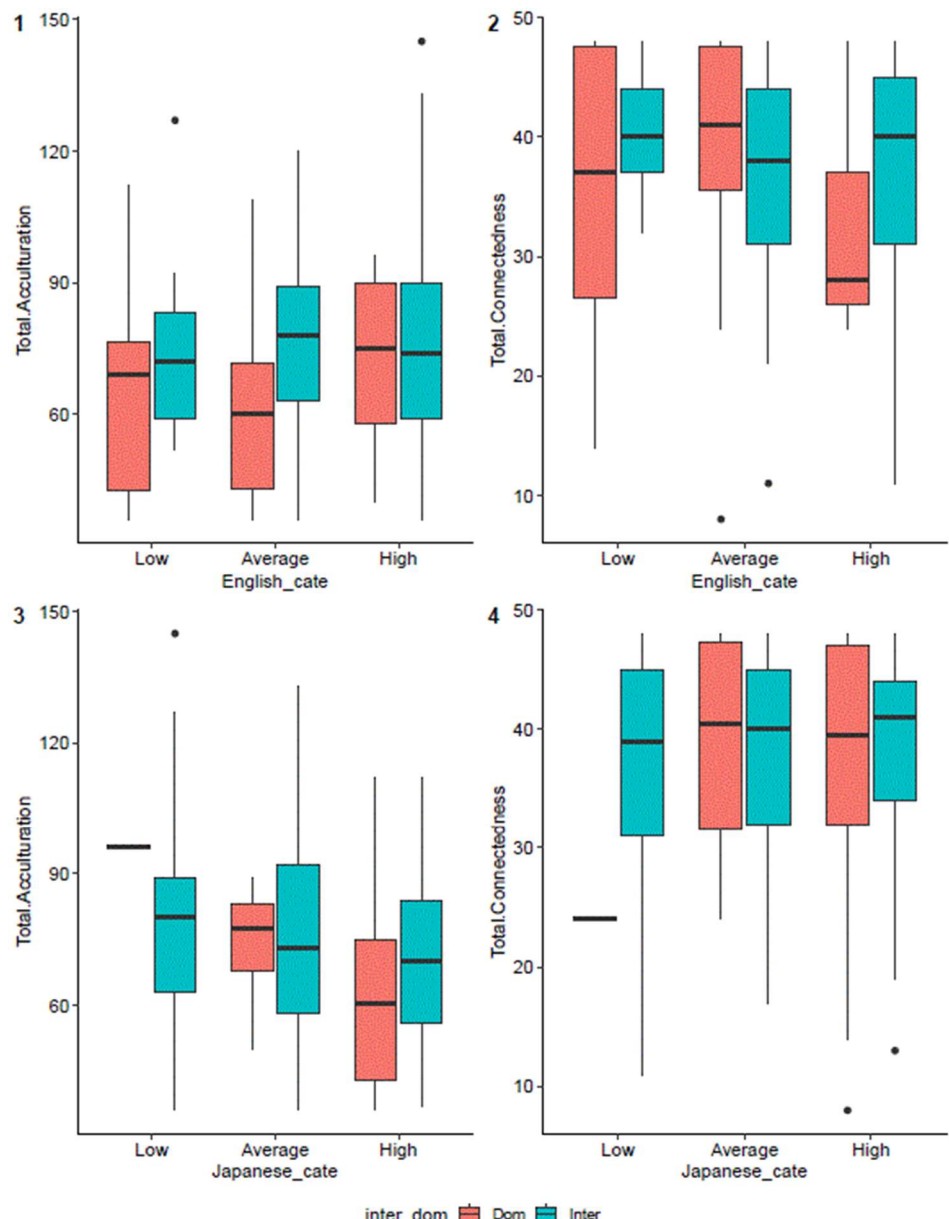

**Figure 3.** Acculturative stress and social connectedness among international and domestic students according to language proficiency.

From Figure 4, we can see that students from different regions with different length of stay in the university have different mental health conditions measured by "ToDep", "ToSC", and "ToAS". Overall, students from most regions experienced the worst mental health conditions during the second and third years of their university life. Distinctively, the level of mental depression and acculturative stress among students from South Asian countries (excluding Japan) increased as they stayed longer, while their sense of connectedness to society decreased over time.

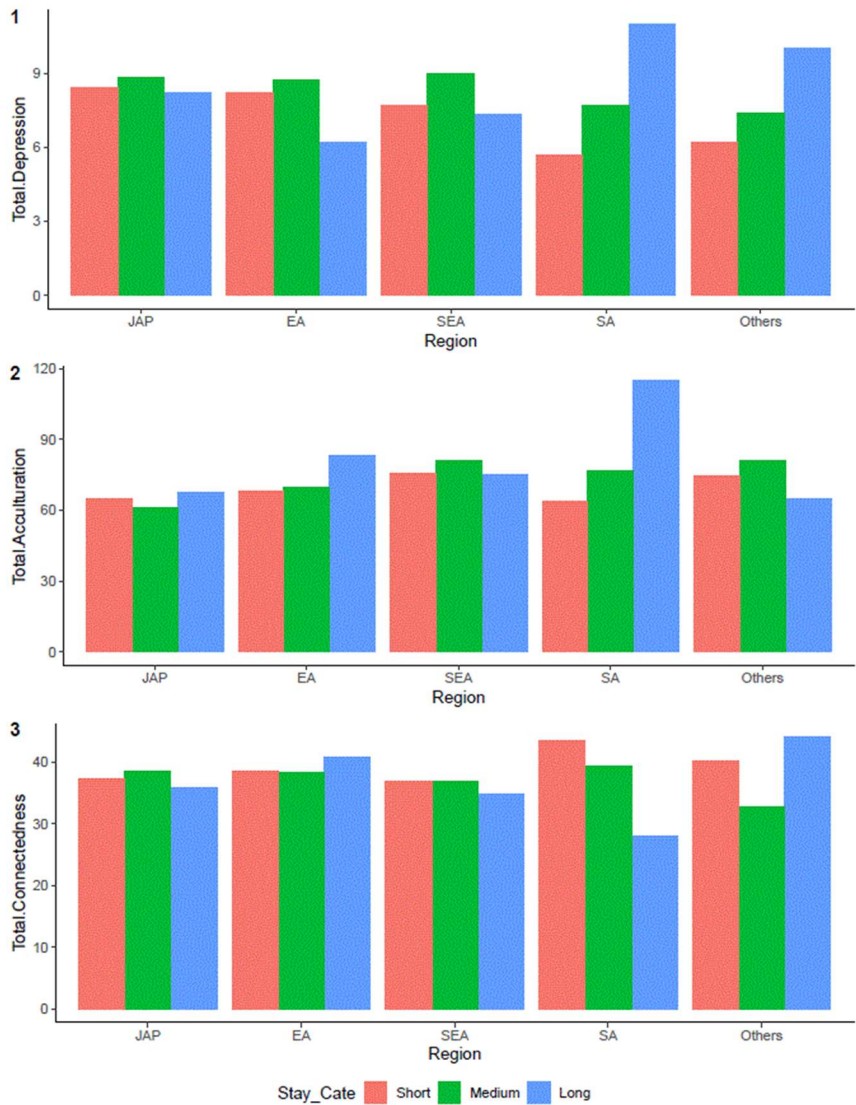

**Figure 4.** Level of depression, acculturative stress, and social connectedness of students from different origins.

### 2.3. Help-Seeking Behaviors

In order to examine the help-seeking behaviors of students, we employed General Health Help-Seeking Questionnaire (GHSQ), in which the contents are easy to modify for the compatible purpose [21]. Based on the GHSQ, ten categorical variables and ten numerical variables were created. However, some students failed to answer whether they were willing to seek help from the Internet.

### 2.4. Potential Research Questions

University students are usually exposed to the risk of mental health problems more than the general population due to the high level of stress concerning futures and employment as well as worrying about studies. According to a systematic review studying publications from 1990 to 2010 on PubMed, PsycINFO, BioMed Central, and Medline, the depression prevalence with 30.6% among university students is higher than the 9% found among the general population [22]. The risk of suffering from mental problems is even greater in international students, because international students face more difficulties due to acculturation [15,16,23,24], and are less accessible to mental health support compared to domestic students due to lack of language proficiency, resources, and fear of discovering a potential negative health condition [13,25,26]. Many studies have been conducted to examine

the prevalence and associated factors of mental health problems among international students and domestic students [27,28], comparative studies between international students and domestic students seem to be a lack in the works of literature. Moreover, besides several studies that have already been done [29–31], the number of articles regarding mental health conditions and help-seeking behaviors seem to be limited in Japan. Therefore, this dataset aims to provide valuable resources to fill in the following gaps in the literature:

1. Lack of studies regarding mental health conditions and help-seeking behaviors in Japan.
2. Lack of comparative studies between domestic and international students in the literature.
3. Lack of studies regarding the effects of acculturation on domestic students.
4. Lack of studies examining the mental health conditions and help-seeking behaviors of international students originating from various countries and regions.

Potential research questions and hypotheses can be examined using this dataset are shown in Table 3.

**Table 3.** Potential research questions and hypotheses.

| | |
|---|---|
| **Research Questions** | What are the socio-demographic determinants of acculturative stress/ social connectedness among international and domestic students? |
| | What kind of help-seeking behaviors can help ease the depressive symptoms and acculturative stress among international and domestic students? |
| | What is the difference in depression prevalence among people from different origins? |
| | What are the determinants of suicidal ideation among international and domestic students? |
| **Hypotheses** | Having a religion is positively correlated with social connectedness among domestic and international students |
| | Having an intimate relationship is negatively correlated with acculturative stress among domestic and international students |
| | Language proficiencies are significantly correlated with depression and acculturative stress among domestic and international students |

The research questions and hypotheses examined in other publications employing this dataset [3,4] are exhibited in Table 4.

**Table 4.** Previously examined research questions and hypotheses.

| | |
|---|---|
| **Research Questions** | What are the socio-demographic determinants of depression among international and domestic students? |
| | What is the difference between the prevalence of depression between international and domestic students? |
| | What are the impacts of acculturative stress on help-seeking behaviors among international and domestic students? |
| **Hypotheses** | Depression and social connectedness are significantly correlated among domestic students and international students |
| | Acculturative stress and social connectedness are significantly correlated among domestic students and international students |
| | Depression and acculturative stress are significantly correlated among domestic and international students |

## 3. Methods

### 3.1. Data Collection

In order to collect the data, we employed a web-based survey methodology because of several reasons: (1) the web-based survey is recently a common methodology which has been validated in medical studies, even with studies related to depressive, anxiety, and stress symptoms [32–34]; and (2) the web-based survey methodology is more cost-effective and has fewer missing values than the paper-based survey methodology [35]. Conversely, the web-based survey methodology also obtains several drawbacks, such as low response rate [35] and bias towards younger and more severe illness possessed population [36]. We initially designed the questionnaire using Google Form, which can secure the confidentiality of the data [37] and is a familiar tool to students. Ritsumeikan APU in Beppu, Japan was chosen as the study site for conducting the survey because of its multicultural environment with an equal proportion of domestic and international students as well as faculties.

The questionnaire employed elements from four standard measurements of mental health and help-seeking behaviors: Patient Health Questionnaire PHQ-9, Acculturation was measured by Acculturative Stress Scale for International Students (ASSIS), Social Connectedness Scale (SCS), and General Health Help-Seeking Questionnaire (GHSQ) (See the dataset). The survey strictly conformed the standard of the WMA Declaration of Helsinki and was approved by the Ethical Committee Board of APU after the internal review. When the committee accepted the questionnaire, the survey period was conducted from October to December 2018. With the easy accessibility for students and efficient data management being prioritized, Google Forms was chosen as the platform to conduct this survey (Sample link: https://forms.gle/zAgByNfHN1LNfnAz6).

The link accessing to the survey was posted on the common space of several classes via the university's internal course management system and Vietnamese community students in the university. Before making the survey available online, we gave a presentation in the distributed classrooms to explain about the purpose, contents, confidentiality and emphasize that the survey was voluntary and self-administered. For the survey posted in the Vietnamese community, we also wrote a post carefully to give a similar explanation with the content of the in-class presentation. The target sample size of the current study was expected to be around 250 students, which was estimated by the population size of 5887 students at APU [38], 95% confidence level, and 6% margin of error.

The survey strictly conformed APU regulations of informing participants about consent text and purpose of the research at the beginning of the survey. Participants can quit the survey any time by either choosing "Not agree" to informed consent or not submitting the survey. Out of 669 students that were asked, two of the authors were able to collect 268 completed answers with the response rate was 40.05%, which was an acceptable number. Moreover, among 268 respondents, some of them failed to report whether they had an intimate partner or not and whether they were willing to seek help from the Internet. Nonetheless, the effect of missing records is negligible, as it does not distort any other variables, but three variables "Intimate", "Internet_bi", and "Internet" should be used with caution.

The data was anonymized using Microsoft Excel and later saved as a CSV file (See the dataset). Eventually, all the analyses were executed in R Studio software (version 3.6).

### 3.2. Data Analysis

The dataset consists of many discrete and continuous variables, which facilitates the use of most of the frequentist techniques, including linear regression analysis and logistic regression analysis. Both mentioned regression analysis techniques were already employed in another study [3]. In the current study, we re-ran only the linear regression analysis employing different statistical software and method taking the logarithm of the dependent variable. Thus, we divided the dataset into two parts: international students and domestic students.

The equations of the current analysis are shown as follows:

$$lnToDep_{inter} = \alpha + \beta_1 ToAS_{inter} + \beta_2 ToSC_{inter} + e \tag{1}$$

$$lnToDep_{dom} = \alpha + \beta_1 ToAS_{dom} + \beta_2 ToSC_{dom} + e \tag{2}$$

where "*lnToDep*" is the logarithm of *ToDep* + 1; $\alpha$ is the constant; $\beta_1$ and $\beta_2$ are coefficients; "*ToAcc*" and "*ToAS*" are variables; and *e* is the white noise. Equations (1) and (2) exhibit the model using the dataset of international and domestic students, respectively. This time, we re-run utilizing R (version 3.6) and taking the logarithm of the dependent variable + 1, which is different from the previous study utilizing Stata (version 15.1) and taking the logarithm of the dependent variable without adding 1. Table 5 shows the estimated results.

**Table 5.** The re-run estimation with "*lnToDep*" being the dependent variable.

|  | International Students | | Domestic Students | |
|---|---|---|---|---|
|  | $\beta$ | *t*-value | $\beta$ | *t*-value |
| **Intercept** | 2.662 | 7.583 *** | 2.339 | 4.06 *** |
| **Social Connectedness** | −0.028 | −4.865 *** | −0.027 | −2.817 ** |
| **Acculturative stress** | 0.005 | 2.264 * | 0.011 | 2.567 * |
| **$R^2$** | 0.23 | | 0.33 | |
| **F (df)** | $F_{(2.198)} = 31.17$ *** | | $F_{(2.64)} = 16.21$ | |

Note: *, ** and *** are statistically significant at 0.05, 0.01 and 0.001, respectively.

The results were estimated using the following R command:

```
# For international students

        > inter$lnDep < −log(inter$ToDep + 1)
        > LinearRegression1 < −lm(lnDep~ToAS+ToSC, data = inter)
        > summary(LinearRegression1)

# For domestic students

        > Dom$lnDep < −log(Dom$ToDep + 1)
        > LinearRegression2 < −lm(lnDep~Total.AS+ToSC, data = Dom)
        > summary(LinearRegression2)
```

The results are also visualized and displayed in Figures 5 and 6, employing the following codes:

```
        > library (ggiraph)
        > library (ggiraphExtra)
        > library (plyr)

# For international students

        > ggplot (inter, aes(y = lnDep, x = ToAS, color = ToSC))
        + geom_point () + stat_smooth(method = "lm", se = FALSE)
        > ggPredict (LinearRegression1,interactive = TRUE)

# For domestic students

        > ggplot (Dom, aes (y = lnDep, x = ToAS, color = ToSC))
        +geom_point() + stat_smooth (method = "lm", se = FALSE)
        ggPredict (LinearRegression2, interactive = TRUE)
```

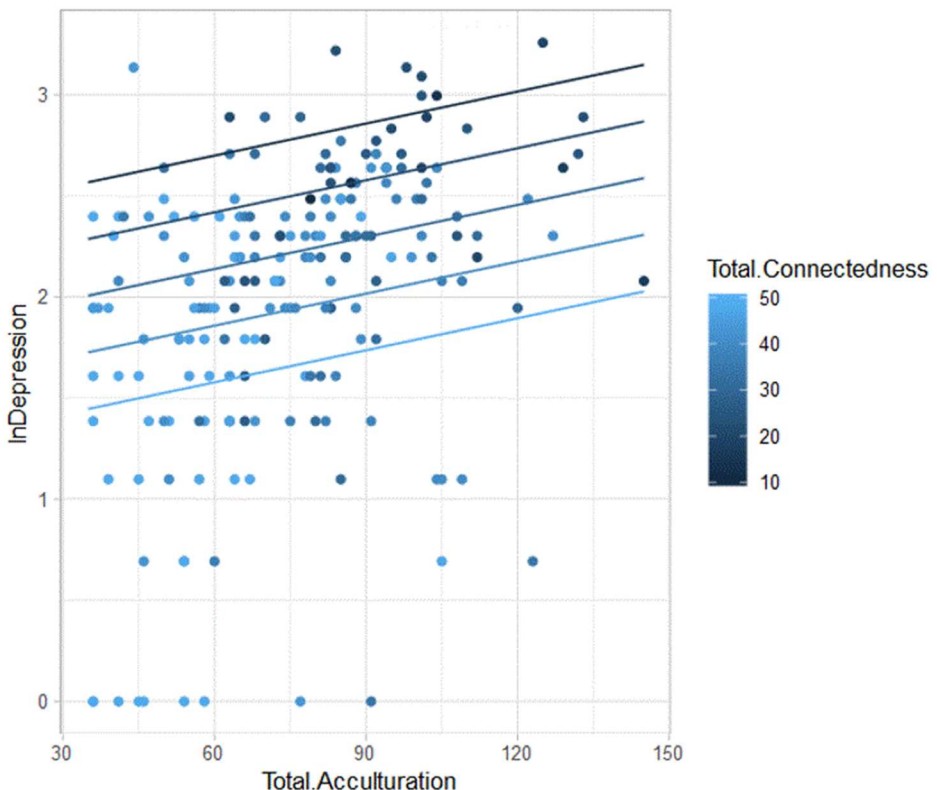

**Figure 5.** The regression line with "ToDep" being the dependent variable using international student dataset.

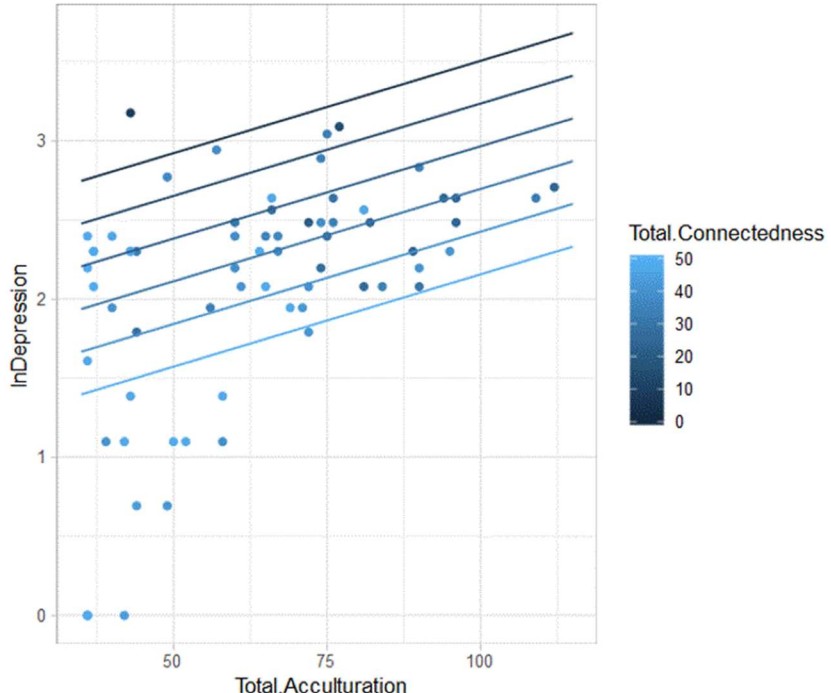

**Figure 6.** The regression line with "ToDep" being the dependent variable using domestic student dataset.

Utilizing different analysis software and a different way to take the logarithm of dependent variables, the re-run analysis indicated a similar result with the previous study despite some differences in coefficients and *p*-value.

## 4. Discussion

This article describes a comprehensive dataset of mental health condition and help-seeking behavior of international and domestic students in an international university in Japan. This dataset has been employed for the publication of two mental health studies which contribute to the literature and policy-making regarding depression, acculturative stress, social connectedness, and help-seeking behaviors [3,4]. Nevertheless, the value of the current dataset has not been completely exploited, and there are still many rooms to make use of it. Thus, open access to this dataset will support further study about the prevalence and associated factors of mental health condition as well as help-seeking behaviors among students in a multicultural environment.

Besides the frequentist approach quantitative analysis, this dataset is also compatible with Bayesian approach analyses. Since the frequentist approach has been doubted for the persistence of "stargazing", p-hacking, and HARKing in social sciences [39,40], Bayesian approach analysis seems to be a potential solution for this [41]. Thus, we recommend this dataset should later be analyzed employing the newest Bayesian analysis tools and software for comparing with existing frequentist approach findings [42,43].

We acknowledge that there are several limitations to this dataset. First, in the sampling procedure, the team slightly modified the questionnaire (ASISS) compared to its original version, it needs to be cautious when comparing the results of this dataset with other studies or dataset. Second, the gender distribution is not equal as the proportion of females (63.43%) is larger than the proportion of males (36.57%). Third, before conducting the survey, the power analysis was not implemented to trace the most suitable sample size of international and domestic students in the current dataset. Another limitation is the imbalance between the number of international and domestic students who participated in the questionnaire. The proportion of international and domestic students were 75% and 25%, respectively. From these points, the result from this dataset should only be used as a reference for policy planners but not be generalized. Moreover, it is recommended that later comparative studies should implement a power analysis prior to the data collection in order to determine the optimal sample size [44]. We hope that in the future, there will be more contribution of open dataset and meta-analysis to the public health studies like the following articles [45–49], as better scientific works can contribute better living standard of the humankind [50].

**Supplementary Materials:** The following are available online at http://www.mdpi.com/2306-5729/4/3/124/s1.

**Author Contributions:** Conceptualization, M.-H.N.; methodology, Q.-H.V., M.-H.N.; software, Q.-H.V.; validation, M.-T.H., Q.-H.V.; formal analysis, M.-H.N., M.-T.H.; data curation, M.-H.N.; writing—original draft preparation, M.-H.N., Q.-Y.T.N.; writing—review and editing, M.-H.N.; M.-T.H.; visualization, Q.-Y.T.N.; supervision, Q.-H.V., M.-T.H.; project administration, Q.-H.V., M.-H.N.

**Funding:** This research received no external funding.

**Acknowledgments:** We would like to show our gratitude to Meirmanov Serik and the Research Office of Ritsumeikan Asia Pacific University for facilitating our research process. We also thank Nguyen To Hong Kong, Ho Manh Tung, Le Tam Tri (Ritsumeikan Asia Pacific University), and La Viet Phuong (Vuong & Associates) for their comments and support.

**Conflicts of Interest:** The authors declare no conflict of interest.

## Table of Acronyms

This table lists the main acronyms used in the article:

| Acronyms | Definition |
| --- | --- |
| APU | Ritsumeikan Asia Pacific University |
| ASSIS | Acculturative Stress Scale for International Students |
| GHSQ | General Health Help-Seeking Questionnaire |
| PHQ-9 | Patient Health Questionnaire |
| SCS | Social Connectedness Scale |

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
