# Peer review of "A Dataset of Students’ Mental Health and Help-Seeking Behaviors in a Multicultural Environment"

_data, 2014_

Round 1

Reviewer 1 Report

The use of the dataset to run other analyses was a good idea. However, more clarification is needed. The purpose of the study was clear. However, the research questions used were unclear. 

Which of the research questions that are listed, apply to this study? Did your analysis answer the research question(s)? How? Please explain.

The research questions are overarching and some have been used in a previous study. Which questions were applied to this present study?

How does the study/data contribute to the literature?

Add a list of acronyms. There is quite a bit of acronym. A list would help simply for the reading for the reader.

Corrections by Line. (Please add the highlighted words).

Line 72: with the mean being 20.87

Line 85: Can you use this data if only 60% of participants responded? What is the value of this feature given the limited response by participants? Why do you think only some participants answered this?

Line 91: Add a period (.) after [5-8].

Line 112: Was the tool used to measure.

Line 114: The questionnaire consists. Change to consisted. Each of them was. Change to were

Line 140: Can this be used given some students "failed to answer the internet question?" What is the significance of this question if all students did not answer it?

Line 144: "According to a systematic review." What review? Please elaborate.

Line 158: Table 3 is labeled "potential research questions and hypotheses. However, I only see one hypothesis. Please revise to reflect hypothesis, not hypotheses or point out the other hypotheses.

Line 174: What is the significance of mentioning Vietnamese community students? Was this a targeted sub-group?

Lines 231-232: Please clarify. Should the data be used or not used as a reference for policy planners?

Author Response

Letter of detailed responses to Reviewer 1

Dear Sir/Madam,

Thank you very much for spending a great amount of time and effort you have put in to reviewing our manuscript. Your detailed comments have helped us improve the quality of our paper.

We have addressed your points in our revised version. Please notice that in the revised paper, the parts that are highlighted in yellow is for correction on the old text, the parts highlighted in green is written anew. Below are our answers to your comments (in italic). Also, the line numbers in the text refer to the revised paper.

The use of the dataset to run other analyses was a good idea. However, more clarification is needed. The purpose of the study was clear. However, the research questions used were unclear. 

Which of the research questions that are listed, apply to this study? Did your analysis answer the research question(s)? How? Please explain.

Thank you for your time and consideration. We will clarify the purpose of the study and the issue relating to the research question below.

This manuscript was submitted as a Data Descriptor, and according to Data’s Aims & Scope (https://www.mdpi.com/journal/data/about), the definition is as follow:

Data Descriptors: the Data Descriptors section publishes descriptions of scientific and scholarly datasets (one dataset per paper). Described datasets need to be publicly deposited prior to publication, preferably under an open license, thus allowing others to re-use the dataset. Small datasets might also be published as supplementary material to the dataset paper in the journal Data. The link to the publicly hosted version of the dataset must be given in the paper. Data descriptors therefore provide easy citability, traceability and accountability of datasets used in scientific research. Research articles published elsewhere based on the data can link back to the data descriptors via a standard reference and DOI number. Data descriptors are published under a CC BY license, thus allowing the reuse of the descriptions in other research papers without copyright infringement.

The purpose of this manuscript is to introduce a dataset of students’ mental health and help-seeking behaviors of international and domestic students in a university located in Japan. Thus, the research questions were listed as potential questions that can be explored using the dataset.

How does the study/data contribute to the literature?

The research questions are overarching and some have been used in a previous study. Which questions were applied to this present study?

Line 158: Table 3 is labeled "potential research questions and hypotheses. However, I only see one hypothesis. Please revise to reflect hypothesis, not hypotheses or point out the other hypotheses.

Thank you for your questions and suggestions. For clarifying which research questions and hypotheses were used and those were not, we have re-structured the expression of research questions and hypotheses. Also, we have added an explanation of the contribution of this dataset to the literature (lines 165 – 178).

Therefore, this dataset aims to provide valuable resources to fill in the following gaps in the literature:

1.         Lack of studies regarding mental health conditions and help-seeking behaviors in Japan.

2.         Lack of comparative studies between domestic and international students in the literature.

3.         Lack of studies regarding effects of acculturation on domestic students.

4.         Lack of studies examining the mental health conditions and help-seeking behaviors of international students originating from various countries and regions.

Potential research questions and hypotheses can be examined using this dataset are shown in Table 3.

Table 3: Potential research questions and hypotheses

Research questions

What are the socio-demographic determinants of acculturative stress/   social connectedness among international and domestic students?

What kind of help-seeking behaviors can help ease the depressive   symptoms and acculturative stress among international and domestic students?

What is the difference of depression prevalence among people from   different origins?

What are determinants of suicidal ideation among international and   domestic students?

Hypotheses

Having a religion is positively correlated with social connectedness   among domestic and international students

Having an intimate relationship is negatively correlated with   acculturative stress among domestic and international students

Language proficiencies are significantly correlated with depression   and acculturative stress among domestic and international students

The research questions and hypotheses examined in other publications employing this dataset [1,2] are exhibited in Table 4.

Table 4: Previously examined research questions and hypotheses

Research questions

What are the socio-demographic determinants of depression among   international and domestic students?

What is the difference between the prevalence of depression between   international and domestic students?

What are the impacts of acculturative stress on help-seeking behaviors   among international and domestic students?

Hypotheses

Depression and social connectedness are significantly correlated among   domestic students and international students

Acculturative stress and social connectedness are significantly   correlated among domestic students and international students

Depression and acculturative stress are significantly correlated among   domestic and international students

Add a list of acronyms. There is quite a bit of acronym. A list would help simply for the reading for the reader.

We have added a list of acronyms for clarification (Line 31-33):

Acronyms

Definition

APU

Ritsumeikan Asia Pacific University

ASSIS

Acculturative Stress Scale for International Students

GHSQ

General Health Help-Seeking Questionnaire

PHQ-9

Patient Health Questionnaire

SCS

Social Connectedness Scale

Line 85: Can you use this data if only 60% of participants responded? What is the value of this feature given the limited response by participants? Why do you think only some participants answered this?

Thank you very much for your question. The survey was distributed through two channels: university’s internal online system and Vietnamese community. Before making the survey online, we gave a presentation in the classrooms to explain about the purpose, contents, confidentiality and emphasize that the survey was voluntary. For the survey distributed in the Vietnamese community, we also wrote a post carefully to explain the purpose, contents, confidentiality and emphasize that doing the survey was voluntary. Moreover, students participating in the survey asked to read the and agree with the consent form based on the university regulations, and in case students did not want to continue, they could cancel the survey at any time by not submitting the survey (the survey was only recorded if students submitted their answers). Through a rigorous process as mentioned, even though the response rate was not high, the dataset can be employed due to the faithfulness of respondents.  Nevertheless, we do agree that we did not explain the impact of missing records clearly, so we have clarified it by adding an explanation of missing records (lines 203 – 206):

Moreover, among 268 respondents, some of them failed to report whether they had an intimate partner or not and whether they were willing to seek help from Internet. Nonetheless, the effect of missing records is negligible, as it does not distort any other variables, but three variables “Intimate”, “Internet_bi”, and “Internet” should be used with caution.

Line 72: with the mean being 20.87

Line 91: Add a period (.) after [5-8].

Line 112: Was the tool used to measure.

Line 114: The questionnaire consists. Change to consisted. Each of them was. Change to were

We have corrected the mistakes. Thank you for your suggestions.

Line 140: Can this be used given some students "failed to answer the internet question?" What is the significance of this question if all students did not answer it?

Thank you for your question. In order to examine the help-seeking behavior of student, we employed the General Health Help-Seeking Questionnaire (GHSQ). One of the advantages of the GHSQ is that new help-seeking sources can be added depending on the target and the environment in which the questionnaire is conducted. Thus, the dataset could still be used, even if all students failed to answer the question whether they were willing to seek help from Internet.

Line 144: "According to a systematic review." What review? Please elaborate.

We have clarified this sentence by adding the reference. Moreover, at the end of the sentence, we cited the review (lines 154 – 155).

“According to a systematic review studying publications from 1990 to 2010 on PubMed, PsycINFO, BioMed Central and Medline, the depression prevalence with 30.6% among university students is higher than the 9% found among the general population [12]”

Line 174: What is the significance of mentioning Vietnamese community students? Was this a targeted sub-group?

Thank you for your question. Vietnamese is not a targeted sub-group. Because all of the authors are Vietnamese, we posted the link to a Facebook group of Vietnamese student community and asked students to fill in the link voluntarily.

Lines 231-232: Please clarify. Should the data be used or not used as a reference for policy planners?

Thank you for pointing this out, we have changed the sentence to clarify the intended meaning (lines 262 – 263):

From these points, the result from this dataset should only be used as a reference for policy planners but not be generalized

We appreciate the hard work and time that you have spent on this manuscript. Your comments and suggestions have helped us improve the quality of our paper. We hope that the revised paper has met your requirements.

Please accept our sincere thanks for your great contributions to the improvement of higher studies on public health and the overall advancement of sciences in the world.

Shall you have further comments, we look forward to hearing from you.

Best regards,

Manh-Toan Ho and Quynh-Yen T. Nguyen

Reviewer 2 Report

A number of factors are considered in coming to this decision, which is not a judgment about the quality of the research presented. Some of these factors include considerations about the general interest of the topic to the broad readership of Data, whether the manuscript would be better suited for a more specialized journal whose readership could more easily appreciate the nuances of the research or whether the work represents only a small advance in understanding of metal-health-relationships.Some other factors are related to possible bias that might occur using a research tool on Google Form.

Minor issues:

Table 3 claims Potential research questions and hypotheses. However, there are only questions, no hypothesis, there.

Author Response

Letter of detailed responses to Reviewer 2

Dear Sir/Madam,

Thank you very much for spending a great amount of time and effort you have put in to reviewing our manuscript. Your detailed comments have helped us improve the quality of our paper.

We have addressed your points in our revised version. Please notice that in the revised paper, the parts that are highlighted in yellow is for correction on the old text, the parts highlighted in green is written anew. Below are our answers to your comments (in italic). Also, the line numbers in the text refer to the revised paper.

A number of factors are considered in coming to this decision, which is not a judgment about the quality of the research presented. Some of these factors include considerations about the general interest of the topic to the broad readership of Data, whether the manuscript would be better suited for a more specialized journal whose readership could more easily appreciate the nuances of the research or whether the work represents only a small advance in understanding of metal-health-relationships. Some other factors are related to possible bias that might occur using a research tool on Google Form.

Thank you for your time and consideration. We will clarify the purpose of the study and the issue relating to the research question below.

This manuscript was submitted as a Data Descriptor, and according to Data’s Aims & Scope (https://www.mdpi.com/journal/data/about), the definition is as follow:

Data Descriptors: the Data Descriptors section publishes descriptions of scientific and scholarly datasets (one dataset per paper). Described datasets need to be publicly deposited prior to publication, preferably under an open license, thus allowing others to re-use the dataset. Small datasets might also be published as supplementary material to the dataset paper in the journal Data. The link to the publicly hosted version of the dataset must be given in the paper. Data descriptors therefore provide easy citability, traceability and accountability of datasets used in scientific research. Research articles published elsewhere based on the data can link back to the data descriptors via a standard reference and DOI number. Data descriptors are published under a CC BY license, thus allowing the reuse of the descriptions in other research papers without copyright infringement.

The purpose of this manuscript is to introduce a dataset of students’ mental health and help-seeking behaviors of international and domestic students in a university located in Japan. Thus, we believe this manuscript meets the requirements to be considered for publications in Data.

In the past, we have also published a Data Descriptors of a dataset that introduced 1024 observations of inpatients’ health care, medical insurance and economic destitution (https://www.mdpi.com/2306-5729/4/2/57) in Data. Only two weeks after the published date, a researcher from Europe had contacted us and expressed his interests in the dataset. Hence, we hope if the dataset that we introduced in this manuscript get to be published in Data, it will be a valuable resource for public uses.

Table 3 claims Potential research questions and hypotheses. However, there are only questions, no hypothesis, there.

Thank you for pointing this out. We have reedited Table 3 according to your suggestion and added Table 4 that shows examined research questions and hypotheses in previous publications for clarity (lines 165 – 178):

Potential research questions and hypotheses can be examined using this dataset are shown in Table 3.

Table 3: Potential research questions and hypotheses

Research questions

What are the socio-demographic determinants of acculturative stress/   social connectedness among international and domestic students?

What kind of help-seeking behaviors can help ease the depressive   symptoms and acculturative stress among international and domestic students?

What is the difference of depression prevalence among people from   different origins?

What are determinants of suicidal ideation among international and   domestic students?

Hypotheses

Having a religion is positively correlated with social connectedness   among domestic and international students

Having an intimate relationship is negatively correlated with   acculturative stress among domestic and international students

Language proficiencies are significantly correlated with depression   and acculturative stress among domestic and international students

The research questions and hypotheses examined in other publications employing this dataset [1,2] are exhibited in Table 4.

Table 4: Previously examined research questions and hypotheses

Research questions

What are the socio-demographic determinants of depression among   international and domestic students?

What is the difference between the prevalence of depression between   international and domestic students?

What are the impacts of acculturative stress on help-seeking behaviors   among international and domestic students?

Hypotheses

Depression and social connectedness are significantly correlated among   domestic students and international students

Acculturative stress and social connectedness are significantly   correlated among domestic students and international students

Depression and acculturative stress are significantly correlated among   domestic and international students

We appreciate the hard work and time that you have spent on this manuscript. Your comments and suggestions have helped us improve the quality of our paper. We hope that the revised paper has met your requirements.

Please accept our sincere thanks for your great contributions to the improvement of higher studies on public health and the overall advancement of sciences in the world.

Shall you have further comments, we look forward to hearing from you.

Best regards,

Letter of detailed responses to Reviewer 1

Dear Sir/Madam,

Thank you very much for spending a great amount of time and effort you have put in to reviewing our manuscript. Your detailed comments have helped us improve the quality of our paper.

We have addressed your points in our revised version. Please notice that in the revised paper, the parts that are highlighted in yellow is for correction on the old text, the parts highlighted in green is written anew. Below are our answers to your comments (in italic). Also, the line numbers in the text refer to the revised paper.

The use of the dataset to run other analyses was a good idea. However, more clarification is needed. The purpose of the study was clear. However, the research questions used were unclear.

Which of the research questions that are listed, apply to this study? Did your analysis answer the research question(s)? How? Please explain.

Thank you for your time and consideration. We will clarify the purpose of the study and the issue relating to the research question below.

This manuscript was submitted as a Data Descriptor, and according to Data’s Aims & Scope (https://www.mdpi.com/journal/data/about), the definition is as follow:

Data Descriptors: the Data Descriptors section publishes descriptions of scientific and scholarly datasets (one dataset per paper). Described datasets need to be publicly deposited prior to publication, preferably under an open license, thus allowing others to re-use the dataset. Small datasets might also be published as supplementary material to the dataset paper in the journal Data. The link to the publicly hosted version of the dataset must be given in the paper. Data descriptors therefore provide easy citability, traceability and accountability of datasets used in scientific research. Research articles published elsewhere based on the data can link back to the data descriptors via a standard reference and DOI number. Data descriptors are published under a CC BY license, thus allowing the reuse of the descriptions in other research papers without copyright infringement.

The purpose of this manuscript is to introduce a dataset of students’ mental health and help-seeking behaviors of international and domestic students in a university located in Japan. Thus, the research questions were listed as potential questions that can be explored using the dataset.

How does the study/data contribute to the literature?

The research questions are overarching and some have been used in a previous study. Which questions were applied to this present study?

Line 158: Table 3 is labeled "potential research questions and hypotheses. However, I only see one hypothesis. Please revise to reflect hypothesis, not hypotheses or point out the other hypotheses.

Thank you for your questions and suggestions. For clarifying which research questions and hypotheses were used and those were not, we have re-structured the expression of research questions and hypotheses. Also, we have added an explanation of the contribution of this dataset to the literature (lines 165 – 178).

“Therefore, this dataset aims to provide valuable resources to fill in the following gaps in the literature:

1.         Lack of studies regarding mental health conditions and help-seeking behaviors in Japan.

2.         Lack of comparative studies between domestic and international students in the literature.

3.         Lack of studies regarding effects of acculturation on domestic students.

4.         Lack of studies examining the mental health conditions and help-seeking behaviors of international students originating from various countries and regions.

Potential research questions and hypotheses can be examined using this dataset are shown in Table 3.

Table 3: Potential research questions and hypotheses

Research questions     What are the socio-demographic determinants of acculturative stress/ social connectedness among international and domestic students?

            What kind of help-seeking behaviors can help ease the depressive symptoms and acculturative stress among international and domestic students?

            What is the difference of depression prevalence among people from different origins?

            What are determinants of suicidal ideation among international and domestic students?

Hypotheses     Having a religion is positively correlated with social connectedness among domestic and international students

            Having an intimate relationship is negatively correlated with acculturative stress among domestic and international students

            Language proficiencies are significantly correlated with depression and acculturative stress among domestic and international students

The research questions and hypotheses examined in other publications employing this dataset [1,2] are exhibited in Table 4.

Table 4: Previously examined research questions and hypotheses

Research questions     What are the socio-demographic determinants of depression among international and domestic students?

            What is the difference between the prevalence of depression between international and domestic students?

            What are the impacts of acculturative stress on help-seeking behaviors among international and domestic students?

Hypotheses     Depression and social connectedness are significantly correlated among domestic students and international students

            Acculturative stress and social connectedness are significantly correlated among domestic students and international students

            Depression and acculturative stress are significantly correlated among domestic and international students

Add a list of acronyms. There is quite a bit of acronym. A list would help simply for the reading for the reader.

We have added a list of acronyms for clarification (Line 31-33):

Acronyms        Definition

APU    Ritsumeikan Asia Pacific University

ASSIS Acculturative Stress Scale for International Students

GHSQ General Health Help-Seeking Questionnaire

PHQ-9 Patient Health Questionnaire

SCS     Social Connectedness Scale

Line 85: Can you use this data if only 60% of participants responded? What is the value of this feature given the limited response by participants? Why do you think only some participants answered this?

Thank you very much for your question. The survey was distributed through two channels: university’s internal online system and Vietnamese community. Before making the survey online, we gave a presentation in the classrooms to explain about the purpose, contents, confidentiality and emphasize that the survey was voluntary. For the survey distributed in the Vietnamese community, we also wrote a post carefully to explain the purpose, contents, confidentiality and emphasize that doing the survey was voluntary. Moreover, students participating in the survey asked to read the and agree with the consent form based on the university regulations, and in case students did not want to continue, they could cancel the survey at any time by not submitting the survey (the survey was only recorded if students submitted their answers). Through a rigorous process as mentioned, even though the response rate was not high, the dataset can be employed due to the faithfulness of respondents.  Nevertheless, we do agree that we did not explain the impact of missing records clearly, so we have clarified it by adding an explanation of missing records (lines 203 – 206):

“Moreover, among 268 respondents, some of them failed to report whether they had an intimate partner or not and whether they were willing to seek help from Internet. Nonetheless, the effect of missing records is negligible, as it does not distort any other variables, but three variables “Intimate”, “Internet_bi”, and “Internet” should be used with caution.”

Line 72: with the mean being 20.87

Line 91: Add a period (.) after [5-8].

Line 112: Was the tool used to measure.

Line 114: The questionnaire consists. Change to consisted. Each of them was. Change to were.

We have corrected the mistakes. Thank you for your suggestions.

Line 140: Can this be used given some students "failed to answer the internet question?" What is the significance of this question if all students did not answer it?

Thank you for your question. In order to examine the help-seeking behavior of student, we employed the General Health Help-Seeking Questionnaire (GHSQ). One of the advantages of the GHSQ is that new help-seeking sources can be added depending on the target and the environment in which the questionnaire is conducted. Thus, the dataset could still be used, even if all students failed to answer the question whether they were willing to seek help from Internet.

Line 144: "According to a systematic review." What review? Please elaborate.

We have clarified this sentence by adding the reference. Moreover, at the end of the sentence, we cited the review (lines 154 – 155).

“According to a systematic review studying publications from 1990 to 2010 on PubMed, PsycINFO, BioMed Central and Medline, the depression prevalence with 30.6% among university students is higher than the 9% found among the general population [12]”

Line 174: What is the significance of mentioning Vietnamese community students? Was this a targeted sub-group?

Thank you for your question. Vietnamese is not a targeted sub-group. Because all of the authors are Vietnamese, we posted the link to a Facebook group of Vietnamese student community and asked students to fill in the link voluntarily.

Lines 231-232: Please clarify. Should the data be used or not used as a reference for policy planners?

Thank you for pointing this out, we have changed the sentence to clarify the intended meaning (lines 262 – 263):

“From these points, the result from this dataset should only be used as a reference for policy planners but not be generalized”

We appreciate the hard work and time that you have spent on this manuscript. Your comments and suggestions have helped us improve the quality of our paper. We hope that the revised paper has met your requirements.

Please accept our sincere thanks for your great contributions to the improvement of higher studies on public health and the overall advancement of sciences in the world.

Shall you have further comments, we look forward to hearing from you.

Best regards,

Manh-Toan Ho and Quynh-Yen T. Nguyen

Reviewer 3 Report

·         Section 1 need to improve in terms of previous studies related to students’ mental health, help-seeking behaviors, and relationship among them. And what is the main contribution/s of this research?

·         The following statement needs a reference/s:

Lines 34-34 “The university is famous for the multicultural environment with 50% of students and faculties being international”

·         Please rewrite a statement between lines 39-41 “The survey was approved by the Ethical…..”

·         In table 1 measuring of the region is not clear “Japan (JAP), South Asia excluding Japan (SA), East Asia (EA), SEA (South East Asia) or other regions (Others)”

·         Inline 44, the authors mentioned the questionnaire was set up on Google form… please mention the link of Google form

·         Sampling process is not enough… for example how they informed the students about their questionnaire…missing data..

·         Please analyze the validity and reliability of the questionnaire

·         Need to explain the type of figure 2. And why the author uses this type of figure to show the correlation between two variables instead of other parametric and non-parametric correlation analysis.

·         From Figure 2 how the authors determined that “Students reporting suffering from major depressive disorder received higher levels of all kinds of acculturative stress than those who have no or other depressive disorders”.

·         Line 185 the authors mentioned logistic regression… please explain which model is logistic regression.

·         If the authors use regression… how about normality test and Multicollinearity test?

Author Response

Letter of detailed responses to Reviewer 3

Dear Sir/Madam,

Thank you very much for spending a great amount of time and effort you have put in to reviewing our manuscript. Your detailed comments have helped us improve the quality of our paper.

We have addressed your points in our revised version. Please notice that in the revised paper, the parts that are highlighted in yellow is for correction on the old text, the parts highlighted in green is written anew. Below are our answers to your comments (in italic). Also, the line numbers in the text refer to the revised paper.

Section 1 need to improve in terms of previous studies related to students’ mental health, help-seeking behaviors, and relationship among them. And what is the main contribution/s of this research?

Thank you very much for your comment. The literature review of this study was discussed in sub-section 2.4. Potential research questions. Based on your recommendation, we have also added a new part describing the contribution of this dataset (lines 152 – 171).

“University students are usually exposed to the risk of mental health problems more than the general population due to the high level of stress concerning futures and employment as well as worrying about studies. According to a systematic review studying publications from 1990 to 2010 on PubMed, PsycINFO, BioMed Central and Medline, the depression prevalence with 30.6% among university students is higher than the 9% found among the general population [14]. The risk of suffering from mental problems is even greater in international students, because international students face more difficulties due to acculturation [15–18], and are less accessible to mental health support compared to domestic students due to lack of language proficiency, resources, and fear of discovering a potential negative health condition [19–21]. Many studies have been conducted to examine the prevalence and associated factors of mental health problems among international students and domestic students [22,23], comparative studies between international students and domestic students seem to be a lack in the works of literature. Moreover, besides several studies that have already been done [24–26], the number of articles regarding mental health conditions and help-seeking behaviors seem to be limited in Japan. Therefore, this dataset aims to provide valuable resources to fill in the following gaps in the literature:

1.         Lack of studies regarding mental health conditions and help-seeking behaviors in Japan.

2.         Lack of comparative studies between domestic and international students in the literature.

3.         Lack of studies regarding effects of acculturation on domestic students.

4.         Lack of studies examining the mental health conditions and help-seeking behaviors of international students originating from various countries and regions.”

The following statement needs a reference/s:

Lines 34-34 “The university is famous for the multicultural environment with 50% of students and faculties being international”

We have provided the references for the statement accordingly (lines 37 – 38):

“The university is famous for the multicultural environment with 50% of students and faculties being international [1,2]

Please rewrite a statement between lines 39-41 “The survey was approved by the Ethical…..”

Thank you for indicating this. We have rewritten the statement as follow (lines 42 – 44):

The author got the permission to conduct the survey from Ethical Committee Board of Ritsumeikan Asia Pacific University (APU) after an internal review. Then, the survey was distributed from October to December 2018.

In table 1 measuring of the region is not clear “Japan (JAP), South Asia excluding Japan (SA), East Asia (EA), SEA (South East Asia) or other regions (Others)”

Thank you for pointing this out. We have rewritten the sentence as follows (line 72):

Japan (JAP), East Asia excluding Japan (EA), South Asia (EA), South East Asia (SEA) or other regions (Others)

Inline 44, the authors mentioned the questionnaire was set up on Google form… please mention the link of Google form

We have provided the link in the manuscript as follow: (Sample link: https://forms.gle/zAgByNfHN1LNfnAz6).

Sampling process is not enough… for example how they informed the students about their questionnaire…missing data..

Thank you for your comment. We have added information regarding how we informed the students about the questionnaire as well as missing records (lines 193 – 206):

The link accessing to the survey was distributed to several classes via the university’s internal course management system and Vietnamese community students in the university. Before making the survey available online, we gave a presentation in the distributed classrooms to explain about the purpose, contents, confidentiality and emphasize that the survey was voluntary. For the survey distributed in the Vietnamese community, we also wrote a post carefully to give a similar explanation with the content of in-class presentation.

The survey strictly conformed APU regulations of informing participants about consent text and purpose of the research at the beginning of the survey. Participants can quit the survey any time by either choosing “Not agree” to informed consent or not submitting the survey. Out of 669 students that were asked, two of the authors were able to collect 268 completed answers, with the response rate was 40.05%. Moreover, among 268 respondents, some of them failed to report whether they had an intimate partner or not and whether they were willing to seek help from Internet. Nonetheless, the effect of missing records is negligible, as it does not distort any other variables, but three variables “Intimate”, “Internet_bi”, and “Internet” should be used with caution.”

Need to explain the type of figure 2. And why the author uses this type of figure to show the correlation between two variables instead of other parametric and non-parametric correlation analysis.

From Figure 2 how the authors determined that “Students reporting suffering from major depressive disorder received higher levels of all kinds of acculturative stress than those who have no or other depressive disorders”.

Thank you for pointing this out. Figure 2 presents a box plot which aims to demonstrate the correlational tendencies between types of depressive symptoms and types of acculturative stress, but not present any parametric nor non-parametric correlation. Perhaps, our explanation made you misunderstood, so we have rewritten the paragraph (lines 112 – 118):

Figure 2 shows the boxplot between types of depressive symptoms and types of acculturative stress among international and domestic students. Students reporting suffering from major depressive disorder seemed to receive higher scores in all kinds of acculturative stress than those who have no or other depressive disorders. Notably, in all cases, the score of international students was relatively more substantial than that of domestic students. Moreover, both international and domestic students without depressive disorders might also be less likely to suffer from acculturative stress.

Please analyze the validity and reliability of the questionnaire

If the authors use regression… how about normality test and Multicollinearity test?

Line 185 the authors mentioned logistic regression… please explain which model is logistic regression.

Thank you very much for your comments. The current manuscript was submitted as a Data Descriptor, and according to Data’s Aims & Scope (https://www.mdpi.com/journal/data/about), the definition is as follow:

Data Descriptors: the Data Descriptors section publishes descriptions of scientific and scholarly datasets (one dataset per paper). Described datasets need to be publicly deposited prior to publication, preferably under an open license, thus allowing others to re-use the dataset. Small datasets might also be published as supplementary material to the dataset paper in the journal Data. The link to the publicly hosted version of the dataset must be given in the paper. Data descriptors therefore provide easy citability, traceability and accountability of datasets used in scientific research. Research articles published elsewhere based on the data can link back to the data descriptors via a standard reference and DOI number. Data descriptors are published under a CC BY license, thus allowing the reuse of the descriptions in other research papers without copyright infringement.

In this paper, our main purpose is to describe the dataset, suggest some types of analyses that can be possibly used for this dataset, and give several simple examples for the readers. Therefore, we would like to simplify the methodologies by not presenting many analysis techniques in the paper.

We did not intend to exhibit the logistic regression analysis in the current study, but only suggested the logistic regression analysis as one of potential analysis, because the logistic regression analysis was already utilized in previous study using this dataset [1]. However, the explanation in the paper might be not clear, so we have rewritten the paragraph (lines 211 – 215).

“The dataset consists of many discrete and continuous variables, which facilitates the use of most of the frequentist techniques, including linear regression analysis and logistic regression analysis. Both mentioned regression analysis techniques were already employed in another study [1]. In the current study, we re-ran only the linear regression analysis employing different statistical software and method taking the logarithm of the dependent variable. Thus, we divided the dataset into two parts: international students and domestic students.”

As for the normality test and multicollinearity test as well as the validity and reliability tests, they were already examined in previous study [1], which employed this dataset. However, based on your recommendation, we have inserted a new part presenting the reliability and validity of the questionnaires (lines 99 – 101, lines 107 – 108, lines 126 – 127):

The measured Cronbach alpha for international and domestic dataset was 0.81 and 0.80 respectively [3], while the validity of the question was confirmed by other mental health studies [10,11].

The internal reliability of the questionnaire was 0.95, and the questionnaire’s validity was also supported by following studies [13–15].

The coefficient alpha of SCS was 0.95, and the validity of the questionnaire was confirmed by [17,18].

We appreciate the hard work and time that you have spent on this manuscript. Your comments and suggestions have helped us improve the quality of our paper. We hope that the revised paper has met your requirements.

Please accept our sincere thanks for your great contributions to the improvement of higher studies on public health and the overall advancement of sciences in the world.

Shall you have further comments, we look forward to hearing from you.

Best regards,

Manh-Toan Ho and Quynh-Yen T. Nguyen

Reviewer 4 Report

It is a very interesting research and data regarding internationals students studying in Japan are important, not many studies are focusing on it. That is why, putting into mirror domestic and international students and presenting more clearly the results should bring a strong point to this research.

Please have pagedown some comments:

- The Introduction section  - should be inserted, in order to connect the purpose of the study  and former and future researches.

 - Please explain why the authors considered not to insert variables regarding student’s home country, gender, academic level, length of stay, languages proficiency, religion, and whether being in an intimate relationship or not are presented and explained in socio-demographic section.

- Please mention the proportion of graduate and undergraduate students. The authors said that "graduate students who participated in the survey are relatively older than undergraduate students." 

- Coded names in Table 1 are not clearly presented

- Please re-write the paragraph - some ideas may be wrongly understood.

lines 118 - 123. "Higher fluency in Japanese results in less acculturative stress among international students, while total connectedness remains the same in all levels of Japanese proficiency. For English proficiency, international students who can use English at a higher level are less likely to suffer from acculturative stress and social disconnectedness. On the contrary, domestic students who are highly fluent in English are negatively affected by acculturation and feel less connected to surrounding people.

- lines 78-84 should split the paragraph.

- line 105. Eliminate re-phrasing the ideas: "Students reporting suffering from major depressive disorder received higher levels of all kinds 106 acculturative stress than those who have no or other depressive disorders. (....)  students without depressive disorders are less likely to suffer from acculturative stress.

- Discussion section is poorly presented, should be improved.

- Data description, Results and Methods sections should be properly presented. Some information from Results section are inserted in Data descriptor or Methods sections.

As mentioned in Limitations sections - the number of internationals students, gender or graduate/undergraduate number of students are important limitations of the study. Please insert data regarding the percentages in order to have a clue.

Author Response

Letter of detailed responses to Reviewer 4

Dear Sir/Madam,

Thank you very much for spending a great amount of time and effort you have put in to reviewing our manuscript. Your detailed comments have helped us improve the quality of our paper.

We have addressed your points in our revised version. Please notice that in the revised paper, the parts that are highlighted in yellow is for correction on the old text, the parts highlighted in green is written anew. Below are our answers to your comments (in italic). Also, the line numbers in the text refer to the revised paper.

- The Introduction section  - should be inserted, in order to connect the purpose of the study  and former and future researches.

Thank you very much for your suggestion. The structure of our paper is based on the template of the journal Data (https://www.mdpi.com/files/word-templates/data-template.dot), which instructs the first section would be 1. Summary: “A short summary of the dataset, methods, background information on why and how the dataset was collected, short description of funded or unfunded research projects that are or will eventually be based on the dataset, and list of publications based on the dataset that were possibly already published.”

So we would like to keep the structure of the paper as it is. However, in order to connect the purpose of the dataset with former and future studies, we have restructured and rewritten the sub-section 2.4 (lines 152 – 178):

“University students are usually exposed to the risk of mental health problems more than the general population due to the high level of stress concerning futures and employment as well as worrying about studies. According to a systematic review studying publications from 1990 to 2010 on PubMed, PsycINFO, BioMed Central and Medline, the depression prevalence with 30.6% among university students is higher than the 9% found among the general population [20]. The risk of suffering from mental problems is even greater in international students, because international students face more difficulties due to acculturation [13,14,21,22], and are less accessible to mental health support compared to domestic students due to lack of language proficiency, resources, and fear of discovering a potential negative health condition [11,23,24]. Many studies have been conducted to examine the prevalence and associated factors of mental health problems among international students and domestic students [25,26], comparative studies between international students and domestic students seem to be a lack in the works of literature. Moreover, besides several studies that have already been done [27–29], the number of articles regarding mental health conditions and help-seeking behaviors seem to be limited in Japan. Therefore, this dataset aims to provide valuable resources to fill in the following gaps in the literature:

1.         Lack of studies regarding mental health conditions and help-seeking behaviors in Japan.

2.         Lack of comparative studies between domestic and international students in the literature.

3.         Lack of studies regarding effects of acculturation on domestic students.

4.         Lack of studies examining the mental health conditions and help-seeking behaviors of international students originating from various countries and regions.

Potential research questions and hypotheses can be examined using this dataset are shown in Table 3.

Table 3: Potential research questions and hypotheses

Research questions

What are the socio-demographic determinants of acculturative stress/   social connectedness among international and domestic students?

What kind of help-seeking behaviors can help ease the depressive   symptoms and acculturative stress among international and domestic students?

What is the difference of depression prevalence among people from   different origins?

What are determinants of suicidal ideation among international and   domestic students?

Hypotheses

Having a religion is positively correlated with social connectedness   among domestic and international students

Having an intimate relationship is negatively correlated with   acculturative stress among domestic and international students

Language proficiencies are significantly correlated with depression   and acculturative stress among domestic and international students

The research questions and hypotheses examined in other publications employing this dataset [1,2] are exhibited in Table 4.

Table 4: Previously examined research questions and hypotheses

Research questions

What are the socio-demographic determinants of depression among   international and domestic students?

What is the difference between the prevalence of depression between   international and domestic students?

What are the impacts of acculturative stress on help-seeking behaviors   among international and domestic students?

Hypotheses

Depression and social connectedness are significantly correlated among   domestic students and international students

Acculturative stress and social connectedness are significantly   correlated among domestic students and international students

Depression and acculturative stress are significantly correlated among   domestic and international students

 - Please explain why the authors considered not to insert variables regarding student’s home country, gender, academic level, length of stay, languages proficiency, religion, and whether being in an intimate relationship or not are presented and explained in socio-demographic section.

- Data description, Results and Methods sections should be properly presented. Some information from Results section are inserted in Data descriptor or Methods sections.

Thank you for addressing this. This manuscript was submitted as a Data Descriptor, and according to Data’s Aims & Scope (https://www.mdpi.com/journal/data/about), the definition is as follow:

Data Descriptors: the Data Descriptors section publishes descriptions of scientific and scholarly datasets (one dataset per paper). Described datasets need to be publicly deposited prior to publication, preferably under an open license, thus allowing others to re-use the dataset. Small datasets might also be published as supplementary material to the dataset paper in the journal Data. The link to the publicly hosted version of the dataset must be given in the paper. Data descriptors therefore provide easy citability, traceability and accountability of datasets used in scientific research. Research articles published elsewhere based on the data can link back to the data descriptors via a standard reference and DOI number. Data descriptors are published under a CC BY license, thus allowing the reuse of the descriptions in other research papers without copyright infringement.

The purpose of this manuscript is to introduce a dataset of students’ mental health and help-seeking behaviors of international and domestic students in an international university located in Japan. Thus, we would like to simplify the methodologies by not presenting many analysis techniques in the paper nor inserting too many variables in the model.

- Please mention the proportion of graduate and undergraduate students. The authors said that "graduate students who participated in the survey are relatively older than undergraduate students." 

Thank you for your comments. We have added the proportion of graduate and undergraduate students. Moreover, the proportion of graduation and undergraduate students were also presented in Table 1 as “Academic” variable (lines 76 – 77).

“The reason for this is because graduate students (7.84%) who participated in the survey are relatively older than undergraduate students (92.16%).”

- Coded names in Table 1 are not clearly presented

Thank you for your comment. We have edited Table 1 to increase its clarity (line 72):

Table 1: Categorical variables

Coded name

Explanation

Unit

Frequency

Proportion

inter_dom

Types of students: International   student (Inter) or domestic student (Dom)

Inter

201

75.00%

Dom

67

25.00%

Region

Regions where students originally   come from: Japan (JAP), East Asia excluding Japan   (EA), South Asia (EA), South East Asia (SEA) or other regions (Others)

JAP

69

25.75%

SA

18

6.72%

EA

48

17.91%

SEA

122

45.52%

Others

11

4.10%

Gender

Gender of students: Male or   Female

Male

98

36.57%

Female

170

63.43%

Academic

Current academic level of   students: Undergraduate (Under) or Graduate School (Grad)

Under

247

92.16%

Grad

21

7.84%

Stay_Cate

How long students have been at   the university: 1 year (Short), 2-3 years (Medium) or at least 4 years (Long)

Short

115

42.91%

Medium

121

45.15%

Long

32

11.94%

Japanese_cate

Self-evaluation scale ranging   from 1 to 5 regarding Japanese proficiency: High (4 to 5), Medium (3) or Low   (1 to 2)

High

87

32.46%

Average

89

33.21%

Low

92

34.33%

English_cate

Self-evaluation scale ranging   from 1 to 5 regarding English proficiency: High (4 to 5), Medium (3) or Low   (1 to 2)

High

166

61.94%

Average

80

29.85%

Low

22

8.21%

Intimate

Whether students have an intimate   partner or not

Yes

103

38.43%

No

157

58.58%

Religion

Whether students are religious or   not

Yes

91

33.96%

No

177

66.04%

Suicide

Whether students have suicidal   Ideation in the last 2 weeks or not (based on a question in PHQ-9)

61

61

22.76%

207

207

77.24%

Dep

Whether students reported to have   depressive symptoms based on PHQ-9 criteria

Yes

96

35.82%

No

172

64.18%

DepType

Types of depressive disorder   based on PHQ-9 criteria: Major depressive disorder (Major), Other depressive   disorder (Other), and no depressive disorder (No)

Major

42

15.67%

Other

54

20.15%

No

172

64.18%

DepSev

The severity of depressive   disorder based on PHQ-9 criteria: Minimal depression (Min), Mild depression   (Mild), Moderate depression (Mod), Moderately severe depression (ModSev),   Severe depression (Sev)

Min

65

24.25%

Mild

107

39.93%

Mod

73

27.24%

ModSev

15

5.60%

Sev

8

2.99%

Partner_bi

Whether students are willing to   seek help from intimate partner when they encounter emotional difficulties

Yes

145

54.10%

No

123

45.90%

Friends_bi

Whether students are willing to   seek help from friends when they encounter emotional difficulties

Yes

128

47.76%

No

140

52.24%

Parents_bi

Whether students are willing to   seek help from parents when they encounter emotional difficulties

Yes

137

51.12%

No

131

48.88%

Relative_bi

Whether students are willing to   seek help from relatives when they encounter emotional difficulties

Yes

66

24.63%

No

202

75.37%

Professional_bi

Whether students are willing to   seek help from professionals when they encounter emotional difficulties

Yes

61

22.76%

No

207

77.24%

Phone_bi

Whether students are willing to   seek help from phone helpline when they encounter emotional difficulties

Yes

30

11.19%

No

238

88.81%

Doctor_bi

Whether students are willing to   seek help from doctor when they encounter emotional difficulties

Yes

46

17.16%

No

222

82.84%

religion_bi

Whether students are willing to   seek help from religious leader when they encounter emotional difficulties

Yes

19

7.09%

No

249

92.91%

Alone_bi

Whether students are willing to   solve problems by themselves

Yes

65

24.25%

No

203

75.75%

Internet_bi

Whether students are willing to   seek help from the Internet when they encounter emotional difficulties

Yes

45

16.79%

No

223

83.21%

Others_bi

Whether students are willing to   seek help from other sources not listed above when they encounter emotional   difficulties

Yes

21

7.84%

No

247

92.16%

- Please re-write the paragraph - some ideas may be wrongly understood.

lines 118 - 123. "Higher fluency in Japanese results in less acculturative stress among international students, while total connectedness remains the same in all levels of Japanese proficiency. For English proficiency, international students who can use English at a higher level are less likely to suffer from acculturative stress and social disconnectedness. On the contrary, domestic students who are highly fluent in English are negatively affected by acculturation and feel less connected to surrounding people.

Thank you very much for your indication. We have rewritten the paragraph as follows (lines 129 – 133):

Among international and domestic students, higher fluency in Japanese might result in less acculturative stress, while the score of total connectedness remained almost similar in all levels of Japanese proficiency. As for English proficiency, there were no clear correlational tendencies between the language proficiency and the acculturative stress as well as the social connectedness.

- lines 78-84 should split the paragraph.

Thank you for your recommendation. We have split the paragraph for higher clarity (lines 81 – 91):

“Other variables such as gender, length of stay, language proficiency and religion were also collected to the dataset. Female participants accounted for 63.43% while this proportion for male students was 36.57%, and until the reported time, most participants had been in this university for 1 to 3 years.

Regarding language proficiency, students were asked to self-evaluate their English and Japanese ability on a scale from 1 to 5. Majority of participants rate themselves 4 or 5, equivalent to high proficiency, for English proficiency (61.94%), while Japanese language evaluation spread equally from low to high proficiency.

Most students reporting to the survey did not consider themselves religious (66.04%). Approximately 60% of students said they did not have an intimate partner (several participants did not respond if they were in an intimate relationship).”

- line 105. Eliminate re-phrasing the ideas: "Students reporting suffering from major depressive disorder received higher levels of all kinds 106 acculturative stress than those who have no or other depressive disorders. (....)  students without depressive disorders are less likely to suffer from acculturative stress.

Thank you for your comment. We have rewritten the paragraph to increase comprehension and coherence (lines 113 – 118):

“Students reporting suffering from major depressive disorder seemed to receive higher scores in all kinds of acculturative stress than those who have no or other depressive disorders. Notably, in all cases, the score of international students was relatively more substantial than that of domestic students. Moreover, both international and domestic students without depressive disorders might also be less likely to suffer from acculturative stress.

- Discussion section is poorly presented, should be improved.

As mentioned in Limitations sections - the number of internationals students, gender or graduate/undergraduate number of students are important limitations of the study. Please insert data regarding the percentages in order to have a clue.

Thank you for your comments. We have added the percentage as you mentioned and made some changes in the Discussion section. Nonetheless, as the manuscript is a data descriptor, we would like not to expand the discussion to specialized contents, but keep the discussion mostly about the value of the dataset and how to exploit that value effectively.

We appreciate the hard work and time that you have spent on this manuscript. Your comments and suggestions have helped us improve the quality of our paper. We hope that the revised paper has met your requirements.

Please accept our sincere thanks for your great contributions to the improvement of higher studies on public health and the overall advancement of sciences in the world.

Shall you have further comments, we look forward to hearing from you.

Best regards,

Manh-Toan Ho and Quynh-Yen T. Nguyen

Round 2

Reviewer 2 Report

I see much improvement in the current version of the manuscript. The authors tried to address most of the raised issues.

I believe the manuscript is improved by including reliability and more descriptives, to understand the data. My main concern is related to the written consent and the use of google docs, as they are recruiting sensitive data (mental health). Therefore, it is important that they state "the research was in accordance with the principles of the Declaration of Helsinki". Furthermore, I have some doubts regarding the use of google as a recruiting tool. I do not know to what extent google can use this information. Moreover, biases might occur. I am really sorry to insist on this, but from an ethical perspective I strongly believe it is crucial.

My second concern is related to the sample size. The manuscript would be significantly improved if this information is provided (e.g., using statistical tools such as G*Power), or at least the authors justified it.

Author Response

Letter of detailed responses to Reviewer 2

Dear Sir/Madam,

Thank you very much for spending a great amount of time and effort you have put in to reviewing our manuscript. Your detailed comments have helped us improve the quality of our paper.

We have addressed your points in our revised version. Please notice that in the revised paper, the parts that are highlighted in yellow is for correction on the old text, the parts highlighted in green is written anew. Below are our answers to your comments (in italic). Also, the line numbers in the text refer to the revised paper.

I see much improvement in the current version of the manuscript. The authors tried to address most of the raised issues.

Thank you for your comment.

I believe the manuscript is improved by including reliability and more descriptives, to understand the data. My main concern is related to the written consent and the use of google docs, as they are recruiting sensitive data (mental health). Therefore, it is important that they state "the research was in accordance with the principles of the Declaration of Helsinki".

Thank you very much for your recommendation. We have included the information that our study strictly conforms the principles of the Declaration of Helsinki:

“The survey strictly conformed the World Medical Association (WMA) Declaration of Helsinki, and was given the permission by the Ethical Committee Board of Ritsumeikan Asia Pacific University (APU) after an internal review.”

Furthermore, I have some doubts regarding the use of google as a recruiting tool. I do not know to what extent google can use this information. Moreover, biases might occur. I am really sorry to insist on this, but from an ethical perspective I strongly believe it is crucial.

Thank you for pointing this out. We have added some points to justify the advantages and disadvantages of the web-based survey methodology. As for the problems using Google Form for survey collection, we believe the confidentiality and anonymity of the information can be maintained due to following reasons: (1) the confidentiality of the data can be secured, because the data is not accessible without the password of the author; and (2) the survey was designed with an intention not to make the respondent identifiable by not asking respondents to sign in their accounts and questions related to their personal details. Besides the information shown in the current dataset, there is only one information can be extracted from the survey which is the time students filled in the survey. Therefore, even the data stored in the cloud database of Google, it is almost impossible that the data can be used to identify anyone.

“In order to collect the data, we employed a web-based survey methodology because of several reasons: (1) the web-based survey is recently a common methodology which has been validated in medical studies, even with studies related to depressive, anxiety, and stress symptoms [30–32]; and (2) the web-based survey methodology is more cost effective and has fewer missing values than the paper-based survey methodology [33]. Conversely, the web-based survey methodology also obtains several drawbacks, such as low response rate [33] and bias towards younger and more severe illness possessed population [34]. We initially designed the questionnaire using Google Form, which can secure the confidentiality of the data [35] and is a familiar tool to students. Ritsumeikan APU in Beppu, Japan was chosen as the study site for conducting the survey because of its multicultural environment with an equal proportion of domestic and international students as well as faculties.

The questionnaire employed elements from four standard measurements of mental health and help-seeking behaviors: Patient Health Questionnaire PHQ-9, Acculturation was measured by Acculturative Stress Scale for International Students (ASSIS), Social Connectedness Scale (SCS), and General Health Help-Seeking Questionnaire (GHSQ) (See the dataset). The survey strictly conformed the standard of the WMA Declaration of Helsinki and was approved by the Ethical Committee Board of APU after the internal review. When the committee accepted the questionnaire, the survey period was conducted from October to December 2018. With the easy accessibility for students and efficient data management being prioritized, Google Forms was chosen as the platform to conduct this survey (Sample link: https://forms.gle/zAgByNfHN1LNfnAz6).”

My second concern is related to the sample size. The manuscript would be significantly improved if this information is provided (e.g., using statistical tools such as G*Power), or at least the authors justified it.

Thank you for your comment. We really appreciate your suggestion using G*Power to justify our sample size. However, as our study has already completed, doing a power analysis will not be plausible. Therefore, we would like to mention your concern related to the sample size as one of our limitation, and give a suggestion for later studies.

“We acknowledge that there are several limitations to this dataset. First, in the sampling procedure, the team slightly modified the questionnaire (ASISS) compared to its original version, it needs to be cautious when comparing the results of this dataset with other studies or dataset. Second, the gender distribution is not equal as the proportion of females (63.43%) is larger than the proportion of males (36.57%). Third, before conducting the survey, the power analysis was not implemented to trace the most suitable sample size of international and domestic students in the current dataset. Another limitation is the imbalance between the number of international and domestic students who participated in the questionnaire. The proportion of international and domestic students were 75% and 25%, respectively. From these points, the result from this dataset should only be used as a reference for policy planners but not be generalized. Moreover, it is recommended that later comparative studies should implement a power analysis prior to the data collection in order to determine the optimal sample size [44]. We hope that in the future, there will be more contribution of open dataset and meta-analysis to the public health studies like the following articles [45–48], as better scientific works can contribute better living standard of the humankind [49].”

Moreover, we have also added a new statement explaining why our sample size is acceptable.

“The target sample size of the current study was expected to be around 250 students, which was estimated by the population size of 5,887 students at APU [38], 95% confidence level, and 6% margin of error.”

We appreciate the hard work and time that you have spent on this manuscript. Your comments and suggestions have helped us improve the quality of our paper. We hope that the revised paper has met your requirements.

Please accept our sincere thanks for your great contributions to the improvement of higher studies on public health and the overall advancement of sciences in the world.

Shall you have further comments, we look forward to hearing from you.

Best regards,

Manh-Toan Ho and Quynh-Yen T. Nguyen

Reviewer 3 Report

The authors amended their manuscript based on all of my comments.

Author Response

Dear Reviewer 3,

Your comments have improved the quality of our manuscript significantly. Thank you for your time and consideration.

Best regards,

The authors

Reviewer 4 Report

I appreciate the corrections and insertions made by the authors.

Please consider the present for publication.

Author Response

Dear Reviewer 4,

Thank you for your time and effort in reviewing our manuscript. The overall quality of the article has improved tremendously.

Best regards,

The authors.

Round 3

Reviewer 2 Report

I see much improvement in the current version of the manuscript being ready for publication.